# Preparation of PVDF/Hyperbranched-Nano-Palygorskite Composite Membrane for Efficient Removal of Heavy Metal Ions

**DOI:** 10.3390/polym11010156

**Published:** 2019-01-17

**Authors:** Xiaoye Zhang, Yingxi Qin, Guifang Zhang, Yiping Zhao, Chao Lv, Xingtian Liu, Li Chen

**Affiliations:** 1State Key Laboratory of Separation Membranes and Membrane Processes, School of Materials Science and Engineering, Tianjin Polytechnic University, Tianjin 300387, China; zhangxiaoye0117@163.com (X.Z.); qinyingxi@163.com (Y.Q.); zgf603@163.com (G.Z.); mariahlv@126.com (C.L.); liuxingtian0413@163.com (X.L.); 2School of Materials Science and Engineering, Tianjin University of Technology, Tianjin 300384, China; tjpuchenlis@163.com

**Keywords:** polymeric composites membrane, palygorskite, PAMAM dendrimers, heavy metal ions adsorption

## Abstract

In this work, three kinds of hyperbranched polyamidoamine-palygorskite (PAMAM-Pal) were designed and synthesized by grafting the first generation polyamidoamine (G1.0 PAMAM), G2.0 PAMAM and G3.0 PAMAM onto Pal surfaces, respectively. Then, these PAMAM-Pals were used as additives to prepare polyvinylidene fluoride (PVDF)/hyperbranched polyamidoamine-palygorskite bicomponent composite membranes. The structures of the composite membranes were characterized by Fourier transform infrared spectroscopy (FTIR), thermo gravimetric analysis (TEM), X-ray photoelectron spectroscopy (XPS), field-emission scanning electronmicroscopy (SEM), atomic force microscope (AFM) and Thermogravimetric analysis (TGA). The adsorption properties of composite membranes to heavy metal ions was studied, and the results found that the maximum adsorption capacities for Cu(II), Ni(II) and Cd(II) could reach 155.19 mg/g, 124.28 mg/g and 125.55 mg/g, respectively, for the PVDF/G3.0 PAMAM-Pal membrane, while only 23.70 mg/g, 17.74 mg/g and 14.87 mg/g could be obtained for unmodified membranes in the same conditions. The high adsorption capacity can be ascribed to the large number of amine-terminated groups, amide groups and carbonyl groups of the composite membrane. The above results indicated that the prepared composite membrane has a high adsorption capacity for heavy metal ions removal in water treatment.

## 1. Introduction

Heavy metal ions, such as copper, cadmium and nickel have been introduced into surface waters by industrial effluents [1]. These heavy metal ions are easily accumulated in the food chain and cannot be naturally degraded. Furthermore, these heavy metal ions are toxic to biosomes even at low concentrations and will do great harm to the ecosystem and human health [2]. Hence, it is necessary to remove heavy metal ions in industrial wastewater. Many treatment techniques have been developed such as chemical precipitation, membrane processes, ion exchange, complexation and adsorption [3,4,5,6,7]. Among these treatment techniques, adsorption is considered as an efficient and promising method for heavy metal ions treatment on account of the comparatively low cost and the ease of operation [8]. Clay minerals such as tobermorite, montmorillonite and palygorskite [9,10,11] are generally selected as effective adsorbent materials in removing heavy metal ions, owing to their large surface areas, high cation exchange capacity, chemical and mechanical stability, and layered structure [12].

Palygorskite is an attractive natural adsorbent because of its low cost, eco-friendliness and very special structure [13]. Palygorskite consists of two Si–O tetrahedron sheets and one (Al, Mg, Fe)–O–OH octahedron sheet and is a kind of crystalline hydrated magnesium aluminum silicate clay mineral with flaky and ribbon structures [14]. Furthermore, it has a highly ordered pore structure and high surface area [15]. In addition, palygorskite surfaces usually have more negative charges which can absorb a certain amount of cations to be electrostatically balance [16]. The above characteristics make palygorskite show great potential for the adsorption of heavy metal ions. However, the general adsorption and ion-exchange are nonspecific which causes the capacity of natural palygorskite to remove heavy metal ions to greatly decline due to the strong competition of coexisteing metal ions for active sites of natural palygorskite [17,18]. In addition, palygorskite is easy to agglomerate in solution which limits the entrance and diffusion of metal ions, leading to the decline of the adsorption [19]. In order to overcome the above drawbacks and then improve the adsorption property of palygorskite, polyamidoamine was selected to modify the natural palygorskite.

Polyamidoamine (PAMAM) dendrimer is an effective surface modification agent [20]. Kaneko et al. reported the grafting of PAMAM onto the surface of colloidal silica particles to successfully prevent their agglomeration in solvents [21]. Zhang et al. prepared well-dispersed epoxy resin/halloysite nanotube composites by functionalization of the HNTs surfaces using PAMAM generation-3(HNTs-G3.0) [22]. The XRD results showed that the grafted PAMAM has no effect on the crystalline structure of HNTs. In addition, PAMAM has an advanced structure which is suitable for improving the capacity of removing heavy metal ions: (1) PAMAM have a large number of amine-terminated groups, amide groups, and carbonyl groups (these functional groups increase exponentially with successive generations), which can form tetradendate complexes with Cu(II) ions [23,24,25,26,27]; (2) the cavity formed by the branches of PAMAM favors its adsorbing and chelating of the heavy metal ions. Even so, recovering the PAMAM and hyperbranched-nano-palygorskite (PAMAM-Pal) particles from industrial effluents is still complex and not cost-effective in practical applications. In order to easily recycle, the modified palygorskite is further considered to be supported on a porous membrane. Due to the high strength, good thermal stability and good resistance to solvents, acids and bases [28,29], poly (vinylidene fluoride) (PVDF) membrane is thus considered to be suitable support.

In this work, a hyperbranched-nano-palygorskite was formed and then used as an additive to prepare a novel PVDF/hyperbranched-nano-palygorskite bicomponent composite membrane for adsorbing heavy metal ions. Experimental results indicated that the prepared membrane shows a high adsorption capacity for heavy metal ions.

## 2. Materials and Methods 

### 2.1. Materials

Reagent-grade ethylenediamine (EDA) and Methyl acrylate (MA) were obtained from Tianjin Kemiou Chemical Reagent Co. (Tianjin, China) and purified by vacuum distillation. PVDF powder was supplied by Solvay Company (Brussels, Belgium). Copper sulfate pentahydrate (CuSO_4_·5H_2_O), cadmium sulfate (3CdSO_4_·8H_2_O), *N*,*N*-Dimethylformamide (DMF), sodium hydroxide (NaOH) and nickel sulfate (NiSO_4_·6H_2_O) were purchased from Tianjin Guangfu Technology Development Co. (Tianjin, China) and were used as received. Reagent-grade γ-aminopropyl triethoxysilane (APTES) was supplied by Tianjin Institute of Fine Chemicals (Tianjin, China). All other chemicals used in this work were of analytical grade and used as received.

### 2.2. Membrane Preparation

The preparation process of hyperbranched-nano-palygorskite can be described using the schematic diagram in Figure 1. PAMAM-Pal were prepared via a multistep in-situ polymerization during the Michael addition and amidation reaction [30]. The preparation of a PAMAM-Pal nano-clay involved a two-step reaction. Before the main two-step reaction, Pal particles were firstly modified by APTES and defined as Pal-G0. The first step was performed as follows: 10.0 g of Pal-G0 was mixed with 200 mL of methanol in a reflux apparatus and treated with ultrasonic vibration for 30 min. Methyl acrylate (MA) was added dropwise to the mixture and refluxed for 1 day at a temperature of 50 °C under a nitrogen atmosphere. After the reaction, using a Soxhlet extractor, the resulting material was repeatedly rinsed with tetrahydrofuran, methylene chloride and anhydrous methanol. The product was dried for 24 h in desiccator at 60 °C, and Pal-G0.5 was obtained. The conditions of the second reaction are based on the first step, 10.0 g of Pal-G0.5 was mixed with 200 mL of methanol in a reflux apparatus and treated with ultrasonic vibration for 30 min. Ethylenediamine (EDA) was added dropwise to the mixture and refluxed for 1 day at a temperature of 25 °C under a nitrogen atmosphere. After the reaction, using a Soxhlet extractor, the resulting material was repeatedly rinsed with tetrahydrofuran, methylene chloride and anhydrous methanol. The product was dried for 24 h in desiccator at 60 °C, and Pal-G1.0 was obtained. Pal-G1.5, Pal-G2.0 and Pal-G3.0 were synthesized and named by repeating the procedure, as shown in Figure 1. A casting solution of pure PVDF was prepared by dissolving 15 wt % PVDF in 85 wt % *N*,*N*-Dimethylformamide (DMF) in a 250 mL flask and refluxed solution at 70 °C for 4 h while stirring with a magnetic stirrer. The bicomponent composite membrane was prepared in a fashion similar to the phase-inversion method of PVDF and DMF and consisted of the basic matrix in the presence of PAMAM-Pal; nanoparticles accounted for 5% of the theoretical weight of PVDF.

### 2.3. Characterization

#### 2.3.1. General Characterization 

The morphologies of the Pal and modified Pal were recorded with a Hitachi H-7650 transmission electron microscope (Hitachi, Kyoto, Japan).

The Pal and modified Pal size and its distribution were measured by dynamic light scattering (DLS) using a zetasizer nano ZS90 (Malvern Instruments, Southborough, UK).

To investigate the chemical compositions of the prepared membrane surfaces, ATR-FTIR spectroscopy was carried out with a Thermo NICOLET8700 FT-IR spectroscope (Thermo Electron Scientific Inc., Waltham, UK) in the range of 4000–400 cm^−1^ using KBr pellets.

Thermogravimetric analysis (TGA) was performed using a TA Instruments Q-5000 IR Analyzer (TA Instruments, New Castle, DE, USA) over a temperature range of 25–800 °C with a scan rate of 10 °C/min in N^2^. The grafting percentage onto the surface of Pal was calculated from the TG analysis as:(1)$$G=m_{2}/m_{1}\times 100\%=\left(m_{3}-m_{4}\right)/m_{4}\times 100\%-\eta$$
where *G* is the grafting percentage onto the surface of Pal; *m*_1_ and *m*_2_ are the weight of adsorbent and grafted material, respectively; *m*_3_ and *m*_4_ are the weight of the samples at 30 °C and 800 °C; η is the weight-loss rate of unmodified Pal.

X-ray photoelectron spectroscopy (XPS) measurements were detected on a GENESIS Model 60S X-ray photoelectron spectrometer (Thermofisher Co., Mahwah, NJ, USA), and the binding energy was corrected with carbon (284.8 eV) as an internal standard.

Scanning Electron Microscope (SEM) analysis was performed with a S-4800 field emission scanning electron microscope (Hitachi, Kyoto, Japan). Wet membrane samples were dried overnight at ambient temperature, fractured after cooling in liquid nitrogen, and sputtered with gold.

Atomic Force Microscope (AFM) analysis was performed with NanoScope IIIa Multimode AFM (Veeco Instruments, Santa Barbara, CA, USA). The surface morphologies and roughnesses of the membranes were obtained by AFM analysis.

Membrane surface was appraised by the apparent static water contact angle at 25 °C with a Data Physics DSA100 Contact angle goniometer (KRÜSS GmbH Co., Hamburg, Germany).

Pure water flux of PVDF membranes was measured by a cross flow filtration system (Laboratory homemade). The measuring process was as follows: 

In the first hour, the membrane was compacted at 0.2 MPa to diminish the compaction effect and the stable flux was obtained. Then the flux was measured at 0.1 MPa. The final steady flux was used as a water flux in this work. The water flux was calculated using the following equation:(2)J=V/t·m
where *J* is the flux for pure water (L/m^2^·h), *V* is the volume for permeate water (L), *m* is the effective membrane area (m^2^), and *t* is the permeation time (h), respectively.

#### 2.3.2. Heavy Metal Ion Adsorption

In this work, Varian 715-ES Inductive Coupled Plasma Emission Spectrometer (Agilent Technologies, Palo Alto, CA, USA) was used for the heavy metal ion adsorption analysis. CuSO_4_·5H_2_O, 3CdSO_4_·8H_2_O and NiSO_4_·6H_2_O as metal ions precursor were dissolved in different distilled water to acquire a concentration of metal of 200 mg/L. The pH of the solution was adjusted with a small amount of NaOH to bring the pH to 7 ± 0.1 (the pH of municipal wastewater). For batch adsorption experiments, a 0.02 m × 0.02 m piece of PVDF membrane was immersed in a cylindrical flask containing 50 mL aqueous Cu(II) solution, and the flask was shaken at 150 rpms in a rotary shaker at room temperature. The experiment was performed in triplicates. The adsorption capacity *Q* (g/m^2^) of the membrane was defined as shown in Formula (3):(3)$$Q=\left(C_{i}V_{i}-C_{f}V_{f}\right)/m$$
where *C_i_*, and *C_f_*, (mg/L) are the initial and final concentrations of the metal ions solution, respectively, *V_i_* and *V_f_* (mL) are the initial and final volume of the aqueous metal ions solution, respectively, and *m* (m^2^) is the membrane area.

Adsorption is a physical–chemical process regarding mass transfers. In order to study the mass transfers mechanism and rate control process of adsorption, the adsorption kinetics equation was used to explore the adsorption process of the membrane [31,32,33,34].

Lagergren-pseudo-first-order equation is in the following form:(4)$$\ln(q_{e}-q_{t})=\ln q_{e}-k_{1}t$$
where *q_t_* and *q_e_* are the amount of adsorption at time *t* (min) and equilibrium (mg/g); *k*_1_ is the rate constant of the equation (L/min).

The pseudo-second-order equation is in the following form:(5)$$t/q_{t}=1/\left(k_{2}q_{e}^{2}\right)+t\times 1/q_{e}$$
where *q_t_* and *q_e_* are the amount of adsorption at time *t* (min) and equilibrium (mg/g); *k*_2_ is the rate constant of the equation (g/mg/min).

## 3. Results

### 3.1. Characterization of the PAMAM-Pal

PAMAM-Pal was prepared by grafting MA and EDA onto Pal-G0 surfaces through the Michael addition and amidation reaction. Due to PAMAM contained characteristic peaks, ATR-FTIR spectral analyses can be used to verify the incorporation of PAMAM into Pal. The ATR-FTIR spectra of Pal and PAMAM-Pal are shown in curve a in Figure 2A. The bands at 783 cm^−1^ and 686 cm^−1^ were attributed to the Si–O–Si vibrations. The band at 974 cm^−1^ was assigned to the Si–O–Si asymmetric stretching vibration [35]. APTES grafted to Pal (Pal-G0 in curve b in Figure 2A) was confirmed by the two new peaks located at 2924 cm^−1^ (C–H bending vibration in the plane) and 1490 cm^−1^ (C–H stretching vibration) [36]. Monomer MA grafted to Pal-G0, Pal-G1.0 (Pal-G0.5 in curve c in Figure 2A, Pal-G1.5 in curve e in Figure 2A) was confirmed by a new band at 1733 cm^−1^ which is assigned to the C=O stretching vibration of the –CO_2_CH_3_ groups. The successful synthesis of Pal-G1.0, Pal-G2.0 and Pal-G3.0 was confirmed by the carbonyl group disappearance and the amidation terminal reorganization conversion to different types of terminal amine and amide groups, as shown in curve d in Figure 2B, curves f and g in Figure 2A: the band at 1554 cm^−1^ can be assigned to the bending vibrations of the C–N–H groups (amide II); the double peak between 3365 cm^−1^ and 3304 cm^−1^ is attributed to the N–H bending vibrations of a unique primary amine. With an increase in the number of generations, the band at 1554 cm^−1^ became stronger, indicating that the surface modification of Pal was successful.

In order to obtain the organic loading of the modified Pal, thermo gravimetric analysis was applied. As shown in curve a in Figure 2B, a three-stage change during heating was observed in the TG result of unmodified Pal, and the total weight loss was approximately 18.72%. This event was associated with the desorption of adsorbed water, crystal water and partially structured water [37]. In comparison, there was an increase in the total weight loss because PAMAM dendrimers grafted on Pal. The total weight loss increased with the generational growth of PAMAM. In fact, the quantity of PAMAM grafted onto the Pal particles increased from 4.94 wt % to 29.46 wt % for G0 to G3.0, as determined by TG analysis. To summarize, TG analysis and ATR-FTIR spectra of the modified Pal indicated that PAMAM bonded to the unmodified Pal surface following the Michael addition reaction and amidation reaction.

More detailed information about the composition of the modified Pal was obtained from XPS spectra which was used to analyze the alteration of the chemical bond evolution upon surface grafting (Figure 3). The spectra of the unmodified Pal and the modified Pal showed characteristic peaks of O1s, Si2p, Mg1s, Al2p, N1s and C1s. As shown in the XPS full-scan spectra curve (Figure 3A), with the generation number increased, the nitrogen content increased, which indicated that the higher-generation PAMAM-Pal (Pal-G2.0, Pal-G3.0) contained more amine groups. Figure 3B–H showed the C1s core level spectra of the unmodified Pal and the modified Pal. The C1s peaks of the unmodified Pal at 285.0 eV and 286.5 eV were mainly attributed to the neutral C–C, C–H and C–O–H. In addition to the neutral C–C and C–H species at 285.0 eV, the C1s signal of Pal-G0 at 286.2 eV indicated the presence of APTES terminal amine groups (C–N). As shown in Figure 3D, Pal-G0.5 contained the new peak at 289.1 eV, corresponds to the carbonyl groups (O–C=O). After multi-step reaction, the Pal-G1.5 (Figure 3F) structure was more complex and was divided into four components corresponding to the neutral C–C, C–H at 285.0 eV, O–C=O at 289.0 eV, O=C–N at 288.0 eV and C–N/C–O at 286.0 eV. There were three fitted peaks at 285.0 eV, 286.0 eV and 288.2 eV, which were assigned to the neutral C–C, C–N/C–O and O=C–N in the spectrum of Pal-G1.0, Pal-G2.0 and Pal-G3.0 and were caused by grafting PAMAM. This result proved the anchoring of –NH_2_ groups at unmodified Pal (shown in Figure 3E,G,H)). According to Figure 3, the high-resolution N1s core-level spectrum of Pal-G1.0, Pal-G2.0 and Pal-G3.0 and the N1s spectrum can be divided into two major components. The stronger peak at approximately 399.8eV was attributed to the amine nitrogen (N–C). The amide nitrogen (O=C–N) occurred at approximately 401.9 eV as a result of the delocalization of the lone pair of electrons of O=C–N [38]. These analyses clearly indicated that PAMAM dendrimers were successfully grafted on the Pal surface.

The morphology of the unmodified Pal and the modified Pal in DMF observed under transmission electron microscopy is shown in Figure 4. The average diameter of the unmodified Pal is 10–70 nm and the length is 500–5000 nm (Figure 4a). This is possibly due to the poor dispersity of the unmodified Pal in DMF leading to their overlapping interleaving and aggregation. Dynamic light scattering (DLS) was further applied to study the dispersity of unmodified Pal. Figure 4 shows a multiple peak distribution of the unmodified Pal’s size, indicating the uneven distribution of the unmodified Pal in DMF. When the hydroxyl groups of the unmodified Pal surface are substituted by APTES (KH-550), the interactions between the unmodified Pal are weakened [39]; thus, a better dispersion of Pal-G0 is obtained(a single peak at about 2007 nm shown in Figure 4). Owing to the primary amine of PAMAM having strong electrophilic ability, the surface tension of PAMAM-Pal was enhanced [25], making Pal-G3.0 well dispersed in DMF (Figure 4c). Furthermore, DLS showed a single peak at about 1223 nm, indicating that Pal-G3.0 in DMF had excellent stability.

### 3.2. Membrane Structure and Performance

The PVDF/hyperbranched-nano-palygorskite bicomponent composite (PVDF/PAMAM-Pal) membrane was prepared via the phase-inversion method, with PAMAM-Pal serving as an additive and a pore-forming agent. The typical ATR-FTIR spectrum of the PVDF membrane was shown in Figure 5, the band at 1179 cm^−1^ and 1400 cm^−1^ attributing to C–H and C–F stretching vibration. The ATR-FTIR spectra of the PVDF membrane and PVDF/PAMAM-Pal membrane are compared in curves a–e in Figure 5. The characteristic FTIR bands of PVDF membrane still existed for PVDF/PAMAM-Pal membranes, which suggested that the structure of PVDF was not destroyed [40]. In addition, PVDF/PAMAM-Pal membranes had new absorption bands at 1650 cm^−1^ and 1545 cm^−1^ due to bending vibrations of N–C=O bonds (amide II) and stretching vibrations of C–N groups [30]. These results confirmed that the modified membranes were successful.

The surface and cross-sectional morphologies of the PVDF/PAMAM-Pal membrane were characterized using SEM. Figure 6a shows that the pure PVDF membrane possessed smooth surface morphologies, no substantial defects and porosity. The unmodified Pal can clearly be observed on the membrane surface due to poor compatibility of the unmodified Pal which trends to aggregation in DMF (Figure 6c). Figure 6c,j,i shows that additional pores, the better dispersion of PAMAM-Pal and higher roughness can clearly be observed on the membrane surface with adding the modified Pal in the matrix. In order to further investigate the effect of adding different generations of modified Pal, we performed an AFM test on the membrane. According to Figure 7, the root mean square roughness (*R_a_*) values of the PVDF, PVDF/Pal, PVDF/Pal-G1.0, PVDF/Pal-G2.0, and PVDF/Pal-G3.0 membrane were 23.7 nm, 34.5 nm, 34.7 nm, 39.5 nm, and 42.2 nm, respectively. The results of the roughness are consistent with SEM images. As shown in Figure 6b,d, the cross-section of the PVDF membrane and the PVDF/Pal membrane exhibited a typical asymmetric structure including a thin skin layer with relatively short finger-like structures. This type of morphology is usually caused by the rapid exchange of a solvent in the casting solution with water. However, the formation of macrovoids was suppressed for the PVDF/PAMAM-Pal membranes. According to Figure 6f,h,j, finger-like pores were observed to grow from the top surface almost to the bottom surface. On the one hand, the addition of PAMAM-Pal increased the viscosity of the polymer solution, which could affect the exchange between DMF and water and subsequently affect the morphology. On the other hand, by improving the compatibility between PAMAM-Pal and DMF, the smaller isolated pores overcame the interfacial tension barrier and gradually interconnected to form the slender finger-like pores throughout the cross section.

The static water contact angle shows that after the addition of PAMAM-Pal, the bicomponent composite membrane exhibited a contact angle that was reduced from 91.0 to 69.6, and a higher hydrophilicity indicated a smaller contact angle, which means that the PVDF/PAMAM-Pal membrane became more hydrophilic. This is mainly due to the increase of surface porosity and the enrichment of hydrophilic PAMAM-Pal on the membrane surface. The external hydrophilic amino of the modified Pal accelerate to the aqueous solution quickly through the membrane surface, which provided a favorable environment for the passage of water penetration. These characteristics correspond with the SEM surface and cross-section results.

Water flux was performed to explore the performance of the PVDF/PAMAM-Pal membranes, and Figure 8 showed water fluxes for the membranes. Water flux of the pure PVDF membrane attained a value of 25.25 L/(m^2^·h), while the value was just 13.22 L/(m^2^·h) for the PVDF/Pal membrane. The reason for this situation was that the unmodified Pal trend to aggregation in DMF, and it blocked the membrane pores. Compared to the PVDF/Pal membrane, the water flux of the PVDF/PAMAM-Pal membrane was higher, especially the PVDF/Pal-G3.0 membrane attained a maximum value of 47.16 L/(m^2^·h). This is possibly due to the better dispersity of the modified Pal in DMF leading to the PVDF/PAMAM-Pal membrane having a higher water flux. Meanwhile, PAMAM-Pal as a pore-forming agent formed the finger-like extensions hole and varying degrees of skin cracks and cortical thinning. Due to hydrophilic structural, PAMAM will be the first to flock to the membrane surface and inside the membrane pores and membrane pore surface when the finger-like extensions hole forming process. The presence of PAMAM in the membrane allows water quickly through the membrane surface, water flux having improved. However, the water flux of the PVDF/Pal-G3.0 membrane was only slightly higher than the PVDF/Pal-G2.0 membrane. This is because the pore-forming effect of the Pal-G3.0 branched structure reaches saturation. The rigid chains of the Pal-G3.0 becomes longer, and tends to distort the membrane structures. The membrane pores are partially blocked, thus preventing a further rise in flux.

### 3.3. Heavy Metal Ions Adsorption and Adsorption Kinetic Model

The adsorption capacities of Pal, PAMAM-Pal, the PVDF membrane and the PVDF/PAMAM-Pal membrane for Cu(II), Ni(II) and Cd(II) were individually investigated at room temperature. The maximum adsorption capacity (*q_m_*) was detected by ICP (shown in Figure 9a,b). The *q_m_* for Cu(II) was just 88.43 mg/g for Pal-G3.0. However, the *q_m_* for Cu(II) was 155.19 mg/g for the PVDF/Pal-G3.0 membrane. According to Figure 9, the PVDF/PAMAM-Pal membrane was favorable for adsorption due to the membrane owning the adsorption capacity by itself and reducing the agglomeration of Pal.

The membranes for heavy metal ions adsorption have a fast adsorption rate, then they gradually slow down and become more balanced, conforming to adsorption kinetics [41]. The adsorption equilibrium curves are shown in Figure 9c–e. For example, the *q_m_* for Cu(II) at 293 K was only 24.8 mg/g for the PVDF membrane; 120 min was substantial to complete the adsorption process. The adsorption capacity of Cu(II) by the newly prepared membrane (the PVDF/Pal-G3.0 membrane) was over 155 mg/g. Thus, in contrast to the PVDF membrane, the PVDF/PAMAM-Pal membrane exhibited very high adsorption capacities for water treatment. The PVDF/PAMAM-Pal membrane not only has a faster rate of adsorption, but also canlast a long time. The PVDF/Pal-G1.0 membrane adsorption equilibrium was 200 min, and the adsorption equilibrium of PVDF/Pal-G3.0 membrane was about 800 min. In addition, the adsorption capacity of the PVDF/PAMAM-Pal membrane was markedly increased with the increasing branching degree of the modified Pal (G0-G3.0). A lot of amino and intramolecular amide of Pal-G3.0 are able to coordinate metal ions, meanwhile the interior cavity of hyperbranched structure also can adsorb metal ions, making the PVDF/PAMAM-Pal membrane has a stronger chelating ability [42]. Since the pore-forming ability of the modified Pal, the metal ions easier into the membrane pores to further action with PAMAM-Pal to improve the adsorption power and durability. Similar results were also observed for Ni(II) and Cd(II), the maximum adsorption capacity for Ni(II) and Cd(II) was over 124.28 mg/g and 125.55 mg/g, respectively. Regarding the other method, a list of representative studies is presented in Table 1. The PVDF/PAMAM-Pal membrane has a much higher adsorption performance compared with the adsorbents. Clearly the new membrane possessed an efficient ability to remove heavy metal ions, resulting in excellent treatment of water.

According to theory, the pseudo-first-order kinetic model is mainly based on the physical adsorption used in the liquid phase adsorption environment, and the pseudo-second-order model is mainly based on the chemical adsorption, including the electronic sharing and electron transfer [53]. The slopes and intercepts of curves by fitting (Figure 10) are used to determine *k* (the constant), *q_e_* (capacity) and *R* (the corresponding linear regression correlation coefficient) (Table 2). The experimental (exp) value of *q_e_* was significantly more than the calculated (cal) value from the pseudo-first-order equation. Meanwhile, the linearized pseudo-second-order kinetics model provided much better *R*^2^ values (all values greater than or equal to 0.95) than those for the pseudo-first-order model, and the calculated values closely approximated the measured values. According to the correlation coefficient, the adsorption mechanism of the hybrid membrane not only includes the electrostatic adsorption, van der Waals forces and the adsorption of surface pores, but also has hydrogen bonding, quiet electricity, amino ligands, chelating adsorption and so on. Therefore, the PVDF/PAMAM-Pal membrane had a higher pseudo-second-order kinetics model correlation coefficient compared to the pseudo-first-order kinetic model.

## 4. Conclusions

A novel organic/inorganic bicomponent composite membrane (PVDF/PAMAM-Pal membrane) was prepared by a phase-inversion method based on the introduction of hyperbranched-nano-palygorskite (PAMAM-Pal) into a PVDF matrix. Because of the addition of the modified Pal additive, macropores in the cross-section of the membrane were suppressed and finger-like pores extended from one membrane surface to the other. The water contact angle of the obtained membrane was clearly improved over that of the pure PVDF membrane; water flux was also increased because of changes in membrane structure and surface morphology. Meanwhile, this type of membrane had a high adsorption capacity for a variety of heavy metal ions such as Cu(II), Cd(II) and Ni(II), and the maximum adsorption capacity for them were over 155.19 mg/g, 124.28 mg/g and 125.55 mg/g, respectively. Therefore, the PVDF/PAMAM-Pal membrane is expected to be applied to the field of water treatment.

## Figures and Tables

**Figure 1 polymers-11-00156-f001:**
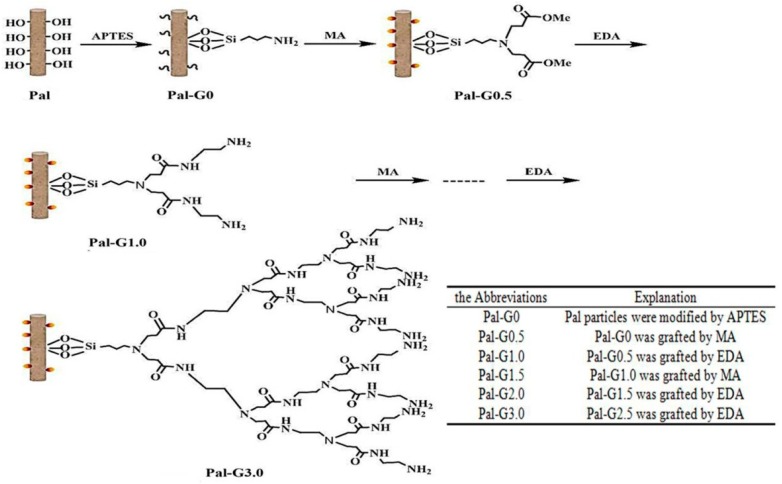
The preparation process of PAMAM-Pal.

**Figure 2 polymers-11-00156-f002:**
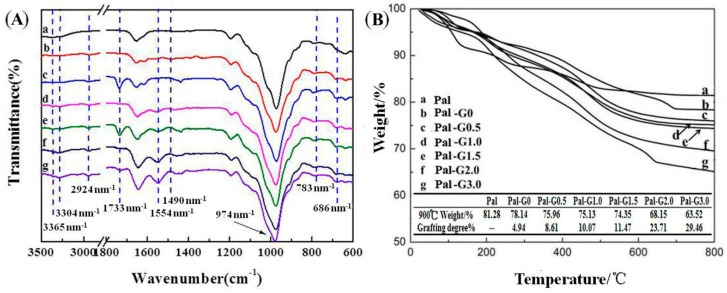
FTIR spectra (**A**) and TG curves (**B**) of modified (a) Pal, (b) Pal-G0, (c) Pal-G0.5, (d) Pal-G1.0, (e) Pal-G1.5, (f) Pal-G2.0 and (g) Pal-G3.0.

**Figure 3 polymers-11-00156-f003:**
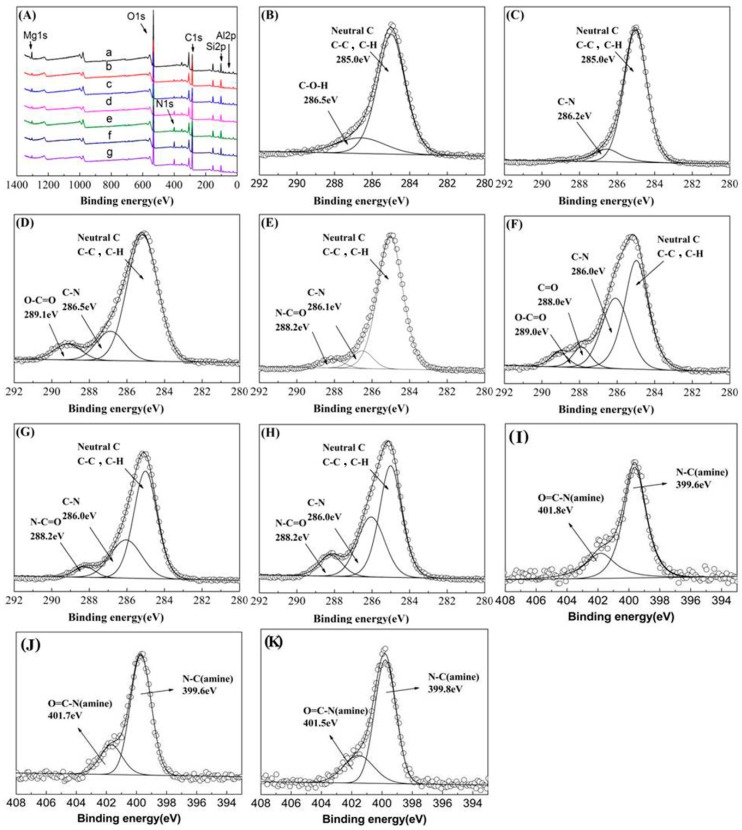
XPS full-scan spectra curve (**A**) of (a) Pal, (b) Pal-G0, (c) Pal-G0.5, (d) Pal-G1.0, (e) Pal-G1.5, (f) Pal-G2.0, and (g) Pal-G3.0; C1s fitting curves of (**B**) Pal, (**C**) Pal-G0, (**D**) Pal-G0.5, (**E**) Pal-G1.0, (**F**) Pal-G1.5, (**G**) Pal-G2.0, and (**H**) Pal-G3.0 and N1s fitting curves of (**I**) Pal-G1.0, (**J**) Pal-G2.0, and (**K**) Pal-G3.0.

**Figure 4 polymers-11-00156-f004:**
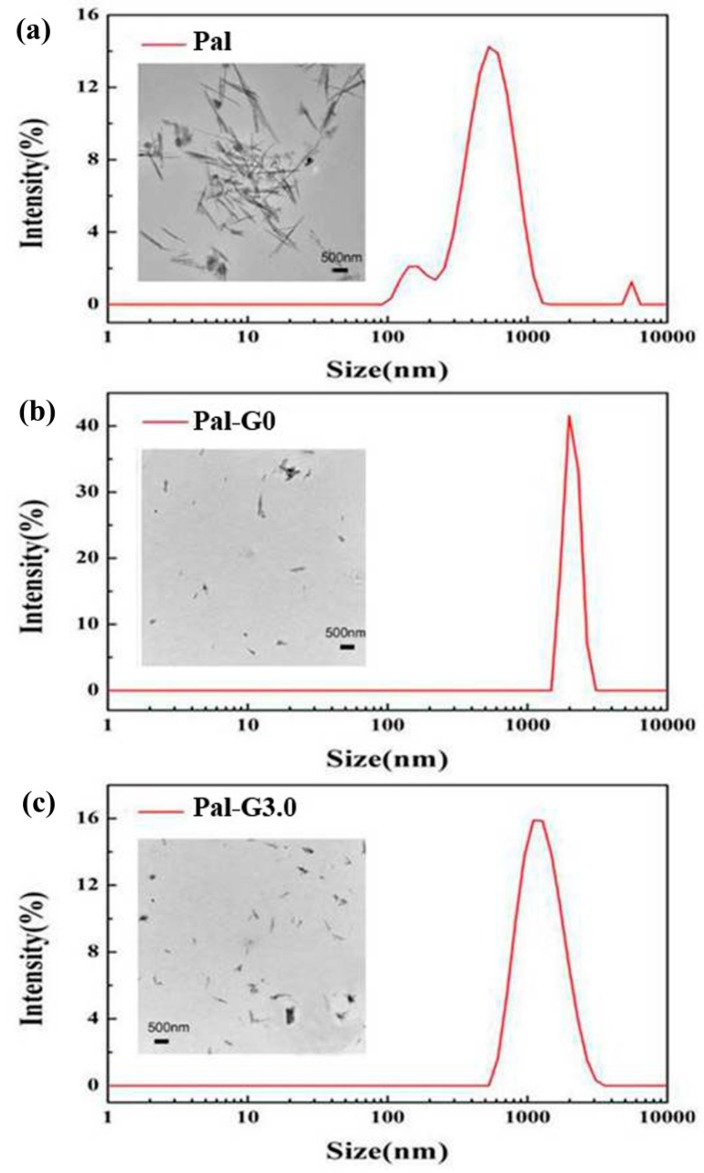
TEM image and Particle size of the (**a**) Pal, (**b**) Pal-G0 and (**c**) Pal-G3.0.

**Figure 5 polymers-11-00156-f005:**
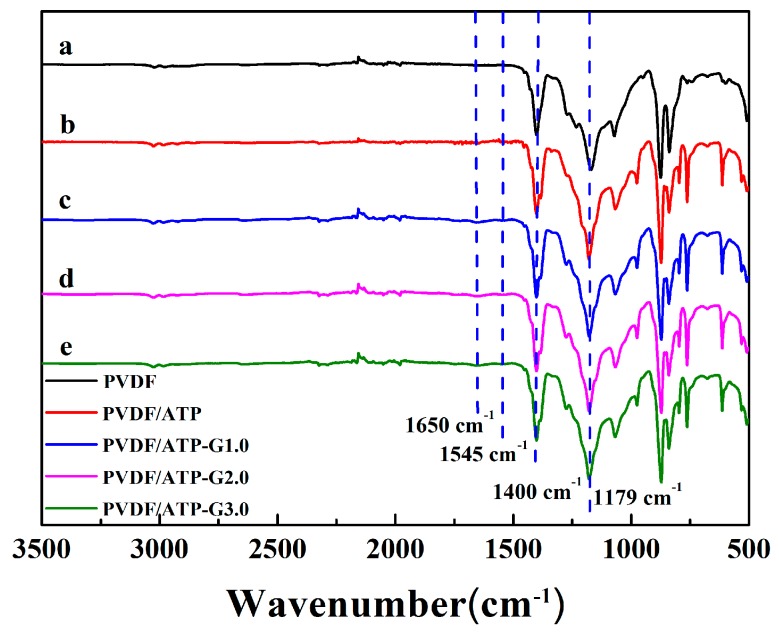
ATR-FTIR spectra for PVDF hybrid membranes.

**Figure 6 polymers-11-00156-f006:**
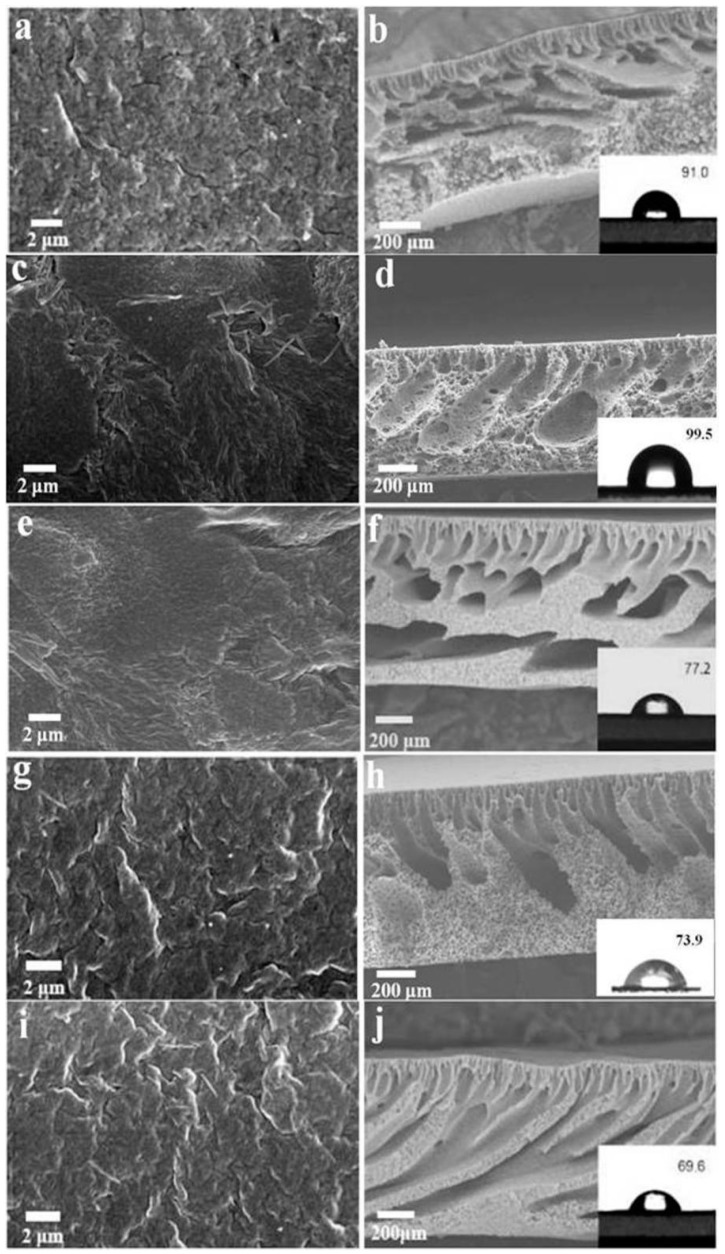
SEM morphologies of hybrid membranes: (**a**,**c**,**e**,**g**,**i**) refer to the top surface of hybrid membranes; (**b**,**d**,**f**,**h**,**j**) refer to the cross-section of hybrid membranes and water contact angle of hybrid membranes. (**a**,**b**) PVDF, (**c**,**d**) PVDF/Pal, (**e**,**f**) PVDF/Pal-G1.0, (**g**,**h**) PVDF/Pal-G2.0, and (**i**,**j**) PVDF/Pal-G3.0.

**Figure 7 polymers-11-00156-f007:**
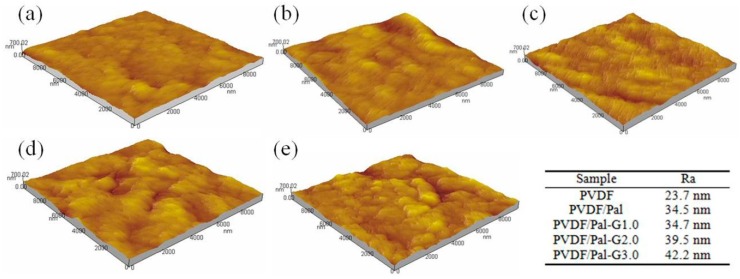
AFM images of hybrid membranes and the roughness (*R_a_*) value is shown under table: (**a**) PVDF, (**b**) PVDF/Pal, (**c**) PVDF/Pal-G1.0, (**d**) PVDF/Pal-G2.0, and (**e**) PVDF/Pal-G3.0.

**Figure 8 polymers-11-00156-f008:**
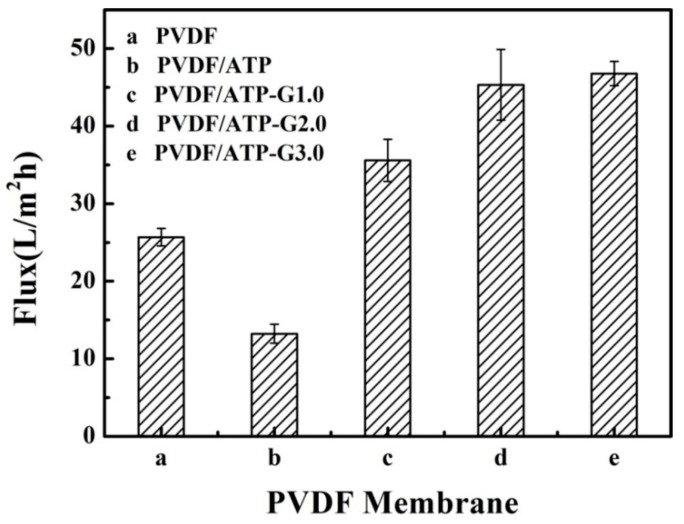
Water flux of hybrid membranes.

**Figure 9 polymers-11-00156-f009:**
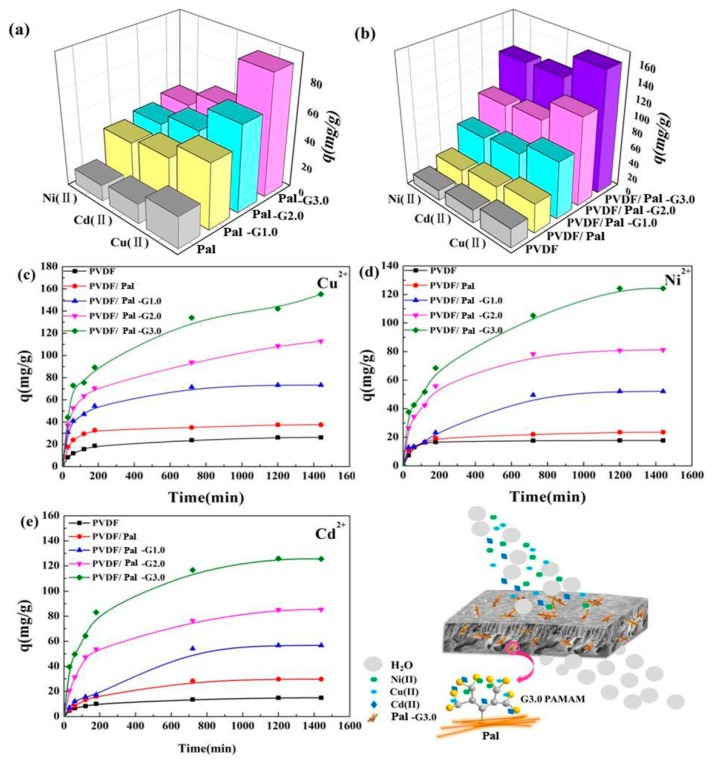
The adsorption capacities and equilibrium adsorption of the amount of heavy metal ions onto hybrid membranes: (**a**,**b**) adsorption capacity, (**c**) Cu(II), (**d**) Ni(II), (**e**) Cd(II) and schematic view of the hybrid membrane structure and adsorption principle.

**Figure 10 polymers-11-00156-f010:**
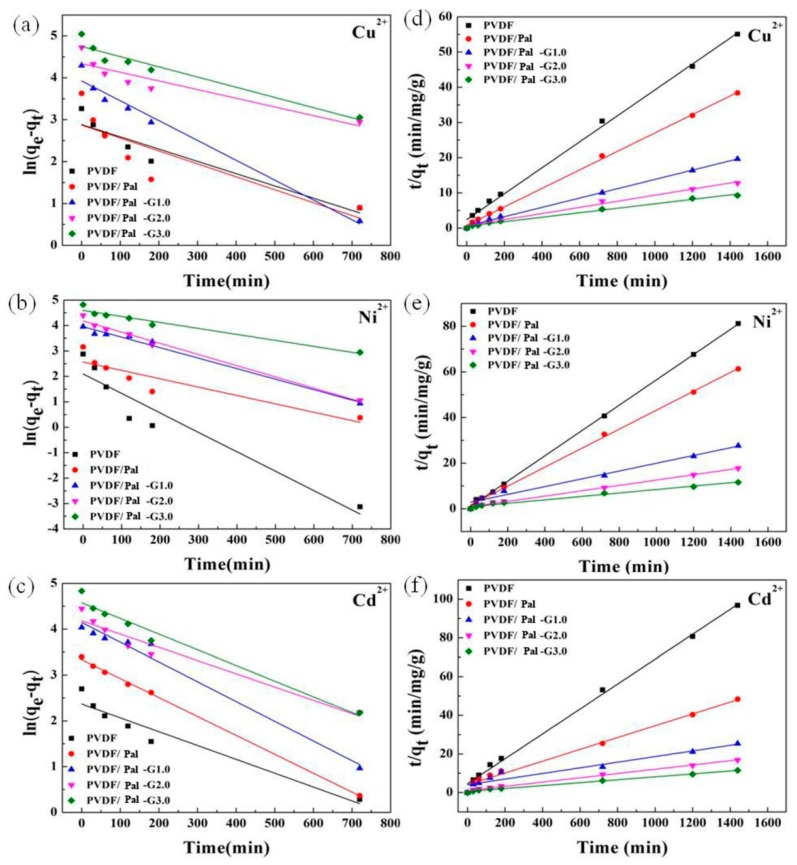
Pseudo-first-order kinetics (**a**–**c**) and Pseudo-second-order kinetics (**d**–**f**).

**Table 1 polymers-11-00156-t001:** Comparison of Maximum Adsorption Capacity (Q) for Cu(II), Ni(II) and Cd(II) with Various Modified Pal Adsorbents.

Adsorbents	C_0_ ^a^ (mg·L^−1^)	Q (mg/g)			Reference
		Cu(II)	Ni(II)	Cd(II)	
porous palygorskite (Pal)/polymer beads	500	25.3	--	32.7	[43]
activated palygorskites	500	--	--	52.99	[44]
Polyacrylic Acid/palygorskite Composite Hydrogels	200	--	72.8	--	[45]
alginate-based palygorskite foams	250	119	--	160	[46]
A novel Ni(II) ion-imprinted polymer	100	--	123.61	--	[47]
struvite/palygorskite	100	--	--	121.14	[48]
palygorskite/carbon nanocomposites	1000	32.32	--	46.72	[49]
CTS-PVA/Pal	210	35.79	--	--	[50]
palygorskite hydroxyapatite composite	120	126.45	--	131.52	[51]
a composite magnetic GO-Pal adsorbent	100	--	190.8	--	[52]
PVDF/hyperbranched-nano-palygorskite composite membrane	200	155.19	124.28	125.55	This work

a: the initial concentration of heavy metal solution.

**Table 2 polymers-11-00156-t002:** Parameters of kinetic models of heavy metal ions adsorption onto hybrid membranes.

Metalions	Samples	*q_e_* (mg/g)	Pseudo-First-Order			Pseudo-Second-Order		
			*q_e_*_1_ (mg/g)	*k*_1_ (min^−^^1^)	*r* ^2^	*q_e_*_2_ (mg/g)	*k*_2_ (g/mg min)	*r* ^2^
Cu(II)	a	24.8547	17.87316	0.0029	0.8834	27.1665	5.64 × 10^−4^	0.9964
	b	37.5432	17.64072	0.0031	0.6403	38.1243	8.78 × 10^−4^	0.9989
	c	73.2668	50.61966	0.0047	0.9704	75.7002	2.89 × 10^−4^	0.9982
	d	112.7836	76.62627	0.0021	0.8211	115.2074	1.13 × 10^−4^	0.9912
	e	155.1950	115.2980	0.0024	0.9188	157.4803	7.35 × 10^−5^	0.9900
Ni(II)	a	12.9696	8.136825	0.0077	0.8915	18.0115	3.05 × 10^−3^	0.9995
	b	23.4972	13.01952	0.0033	0.7964	24.0211	1.09 × 10^−3^	0.9986
	c	52.1184	52.18734	0.0041	0.9870	58.5480	1.02 × 10^−4^	0.9688
	d	81.2291	65.37566	0.0044	0.9861	85.3242	1.63 × 10^−4^	0.9957
	e	124.2819	98.64721	0.0024	0.9503	132.6260	6.54 × 10^−5^	0.9853
Cd(II)	a	17.1907	10.69846	0.0030	0.9238	15.5400	8.91 × 10^−4^	0.9958
	b	29.7817	28.11463	0.0041	0.9986	32.7118	2.32 × 10^−4^	0.9876
	c	56.7012	63.24145	0.0043	0.9749	68.6342	5.21 × 10^−5^	0.9208
	d	85.3710	65.50392	0.0029	0.9355	90.2527	1.22 × 10^−4^	0.9939
	e	125.5581	97.09792	0.0034	0.9658	132.4503	9.39 × 10^−5^	0.9948

a: the PVDF membrane; b: the PVDF/Pal membrane; c: the PVDF/Pal-G1.0 membrane; d: the PVDF/Pal-G2.0 membrane; e: the PVDF/Pal-G3.0 membrane.

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
