# Peer review of "Preparation of PVDF/Hyperbranched-Nano-Palygorskite Composite Membrane for Efficient Removal of Heavy Metal Ions"

_polymers, 2019, doi:10.3390/polym11010156_

Round 1
Reviewer 1 Report
In this manuscript, various PAMAM-Pal was designed for the adsorption of heavy atoms. The work is interesting for the literature. However, a minor revision is necessary:
1. Language needs proofreading.
2. There is various type of Pal such as G1.0-0.5-1.5-2-3-5 (Figure 2). However, the preparation of each has not been defined. It is better to prepare a table for the abbreviations and their explanation.
3. Page 2, line 58: “PAMAM dendrimer is an effective surface treating agent.” Which kind of surface treatment?
4. Page 2, line 89: it has been mentioned as PAMAM-Pal nanofiber as first and last time. Is it nanofiber form? If it is, it is not visible via SEM. What is the diameter?
5. The preparation methods for the each Pal –G should be extended more in the membrane preparation.
6. Page 3, Line 99, it has been mentioned PAMAM-Pal G3.0 was prepared under optimum condition, what is this condition?
7. Page 4, Line 143: The metal ions precursor was dissolved in deionized water. What is the amount of each precursor?
8. The text and the numbers in the Figure 2 is not visible, it should be bigger font size.
9. What is Pal G1.5 and Pal-G2.0?
10. Figure 5(b) should be mentioned before Figure 6, or figure 5 should be given as separate as Figure 7.
11. Table 2, what is a, b, c, d, and e?
Author Response
Thank you for your letter and for the reviewer’s comments concerning our manuscript entitled “Preparation of PVDF/hyperbranched-nano-palygorskite composite membrane for efficient removal of heavy metal ions” (polymers-422019). Those comments are all valuable and very helpful for revising and improving our paper, as well as the important guiding significance to our researches. We have studied these comments carefully and have made correction which we hope to meet with your approval.
Please find our detailed responses in the attached PDF.

Reviewer 2 Report
The manuscript by Zhang et al. reported a new type of composite membrane based on PVDF and PAMAM modified hyperbranched-nano-palygorskite. Adding the filler enhanced the surface hydrophilicity, water flux, membrane morphology and its adsorption capacity towards heavy metal ions. Overall, the manuscript is written well, and the conclusions are supported with characterization data. I would recommend publishing the manuscript after addressing the following relatively minor issues:
1. In the morphology section, authors claimed that the roughness of the membranes is changed by adding the filler. However, it is hard to follow up the changes in surface roughness using the SEM images. I would suggest authors provide AFM images and calculate the membranes roughness according to the AFM data.
2. In Fig. 5(a), the FTIR peaks of nanocomposite membranes have overlap, and as a result, most of the peaks have mixed up. Please adjust the figure.
3. Please provide the molecular weight cut-off of the prepared membranes. No rejection data is provided either for pure PVDF membrane or the composite structures.
Author Response
Thank you for your letter and for the reviewer’s comments concerning our manuscript entitled “Preparation of PVDF/hyperbranched-nano-palygorskite composite membrane for efficient removal of heavy metal ions” (polymers-422019). Those comments are all valuable and very helpful for revising and improving our paper, as well as the important guiding significance to our researches. We have studied these comments carefully and have made correction which we hope to meet with your approval.
Please find our detailed responses in the attached
